# Proposal for dark exciton based chemical sensors

Maja Feierabend[1,2], Gunnar Berghäuser[1,2], Andreas Knorr[2] & Ermin Malic[1]

The rapidly increasing use of sensors throughout different research disciplines and the demand for more efficient devices with less power consumption depends critically on the emergence of new sensor materials and novel sensor concepts. Atomically thin transition metal dichalcogenides have a huge potential for sensor development within a wide range of applications. Their optimal surface-to-volume ratio combined with strong light–matter interaction results in a high sensitivity to changes in their surroundings. Here, we present a highly efficient sensing mechanism to detect molecules based on dark excitons in these materials. We show that the presence of molecules with a dipole moment transforms dark states into bright excitons, resulting in an additional pronounced peak in easy accessible optical spectra. This effect exhibits a huge potential for sensor applications, since it offers an unambiguous optical fingerprint for the detection of molecules—in contrast to common sensing schemes relying on small peak shifts and intensity changes.

[1] Department of Physics, Chalmers University of Technology, SE-412 96 Gothenburg, Sweden. [2] Institut für Theoretische Physik, Nichtlineare Optik und Quantenelektronik, Technische Universität Berlin, Hardenbergstraße 36, 10623 Berlin, Germany. Correspondence and requests for materials should be addressed to M.F. (email: maja.feierabend@chalmers.se).

**D**uring the past decades, sensors have become a focus of attention in development and research due to increasing quality demands and advancement in signal processing[1–4]. The requirements for an optimal sensor are acceptable cost and error rate, fast response time and diverse applicability, for example, in a broad temperature and energy range. To be able to meet these requirements, new sensor materials and novel sensor concepts are needed. One class of promising materials are atomically thin transition metal dichalcogenides (TMDs). They are characterized by an optimal surface-to-volume ratio, giving rise to a high sensitivity to changes in their environment. Furthermore, due to a direct band gap and an extraordinarily strong Coulomb interaction, TMDs exhibit efficient light–matter coupling and tightly bound excitons[5–8] that can be exploited for sensing based on optical read-out. Beside optically accessible bright excitons located in the K valley, TMDs also show a variety of optically forbidden (dark) excitons that either exhibit a non-vanishing angular or centre-of-mass momentum[9–11]. One especially interesting dark state is the KΛ exciton, where the hole is located at the K and the electron at the Λ valley (see Fig. 1a). Recent theoretical and experimental studies[12–15] have demonstrated that there are dark excitons located energetically below the bright KK exciton in tungsten-based TMDs (see Fig. 1b). This can be ascribed to two effects: (1) different spin degeneracy in the conduction band for tungsten- and molybdenum-based TMDs and (2) different excitonic binding energy for K and Λ valley due to their specific effective mass. Here, we show that an efficient coupling between dark and bright TMD excitons and non-covalently attached molecules with a strong dipole moment can turn the dark KΛ exciton bright resulting in an additional peak in optical spectra (see Fig. 1c). This effect exhibits a huge potential for sensor applications, since it offers a clear optical fingerprint for the detection of molecules.

## Results

**Theoretical approach.** Our theoretical approach is based on the Wannier equation providing access to eigenvalues and eigenfunctions for bright and dark excitons and the Bloch equation for the microscopic polarization giving access to excitonic optical spectra including the interaction with externally attached molecules. Solving these equations, we show that the dark KΛ exciton becomes visible through the exciton-molecule coupling at a broad range of temperatures. We also demonstrate how the energetic position of the dark exciton peak can be controlled by the applied substrate and the characteristics of the molecules. In a limiting case, we present an analytic solution offering valuable insights into the underlying elementary processes.

The main goal of our study is to investigate the question of whether dark excitons can be exploited for sensing of molecules. To answer this question, we calculate the excitonic absorption spectrum of tungsten disulfide (WS$_2$), an exemplary TMD material that has been functionalized with merocyanine molecules representing a class of photoactive molecules with a strong dipole moment[16]. To have access to the absorption spectrum, we determine the temporal evolution of the microscopic polarization $p^{vc}_{\mathbf{k_1 k_2}}(t)$. This microscopic quantity is a measure for optically induced transitions from the state $(v, \mathbf{k_1})$ to the state $(c, \mathbf{k_2})$ that are characterized by the electronic momentum $\mathbf{k}_i$ and the band index $\lambda_i = (v, c)$ denoting the valence and the conduction band, respectively[17].

Since excitonic effects are known to dominate optical properties of TMDs[5,6,18], we project the microscopic polarization into an excitonic basis[19] $p^{vc}_{\mathbf{k_1 k_2}}(t) \rightarrow p^{vc}_{\mathbf{qQ}}(t) = \sum_\mu \varphi^\mu_{\mathbf{q}} p^\mu_{\mathbf{Q}}(t)$ with excitonic eigenfunctions $\varphi^\mu_{\mathbf{q}}$ and the index $\mu$ representing the excitonic state, for example, KK or KΛ exciton (see Fig. 1b).

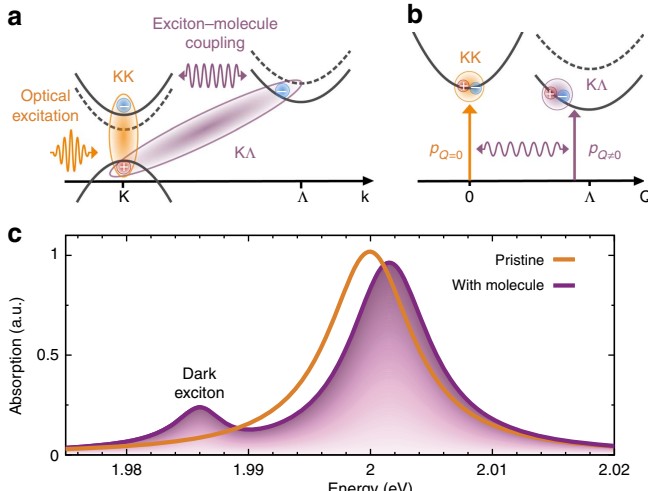

**Figure 1 | Excitonic dispersion and appearance of dark excitons.** (**a**) Electronic dispersion of TMDs exhibits high symmetry K and Λ valleys. An optical pulse excites direct bright KK excitons, where both electron and hole are located at the K valley. These excitons interact with optically inaccessible indirect KΛ dark excitonic states (hole located in K and electron in Λ valley) via coupling with molecules. (**b**) Excitonic dispersion with bright (dark) excitons exhibiting a centre-of-mass momentum $Q = 0$ ($Q \neq 0$). Note that the dark exciton lies below (above, dashed lines) the bright exciton in tungsten (molybdenum)-based TMDs. (**c**) Excitonic absorption spectrum of pristine and merocyanine-functionalized WS$_2$ on a silicon dioxide substrate at 77 K. Efficient exciton–molecule coupling gives rise to an additional peak that can be ascribed to the dark KΛ exciton.

Furthermore, we introduce centre-of-mass and relative momenta **Q** and **q**, where $\mathbf{Q} = \mathbf{k_2} - \mathbf{k_1}$ and $\mathbf{q} = \frac{m_h}{M}\mathbf{k_1} + \frac{m_e}{M}\mathbf{k_2}$ with the electron (hole) mass $m_{e(h)}$ and the total mass $M = m_e + m_h$. The separation ansatz enables us to decouple the relative from the centre-of-mass motion. For the relative coordinate, we solve the Wannier equations[6,20–22] corresponding to the excitonic eigenvalue problem offering access to excitonic eigenfunctions $\varphi^\mu_{\mathbf{q}}$ and eigenenergies $\varepsilon_\mu$. Our approach is consistent with previous studies[6,14] that revealed strongly bound excitons with binding energies of ∼0.5 eV and excitonic linewidths in the range of tens of meV—in good agreement with recent experimental findings[5,7]. Applying the effective mass approximation, our investigations are limited to processes around the high-symmetry points in the Brillouin zone. Furthermore, we focus on the exciton–molecule interaction in this work and neglect the impact of exciton–exciton and exciton-disorder processes.

To obtain the temporal evolution of the excitonic microscopic polarization $p^\mu_{\mathbf{Q}}(t)$, we solve the TMD Bloch equation[6] explicitly including the carrier–light, carrier–carrier and carrier–molecule interaction (see the Methods section for more details on the theoretical approach). Note that only the bright KK exciton with a zero centre-of-mass momentum can be optically excited, that is, only the microscopic polarization $p^K_{\mathbf{Q}=0}$ is driven by the vector potential (see the appearing Kronecker in equation (4) in the Methods section). The dark KΛ exciton is indirectly driven by the exciton–molecule interaction $G^{\mu\nu}_{\mathbf{Qk}}$ that depends on the overlap of the involved excitonic wave functions and the molecule characteristics, such as molecular coverage, dipole moment and the distance between the TMD layer and the layer of attached molecules. One can imagine a molecular lattice in the real space, where the corresponding lattice constant determines the centre-of-mass moment that can be provided by the molecules for indirect scattering within the actual TMD lattice.

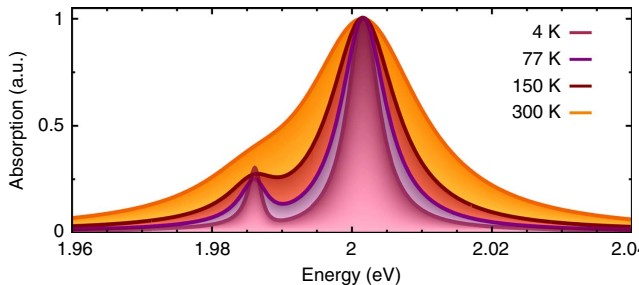

**Figure 2 | Temperature dependence.** While at lower temperatures a clearly visible additional peak due to dark excitons can be observed in absorption spectra of merocyanine-functionalized WS$_2$, at room temperature only a shoulder can be seen due to the increased linewidth of excitonic resonances.

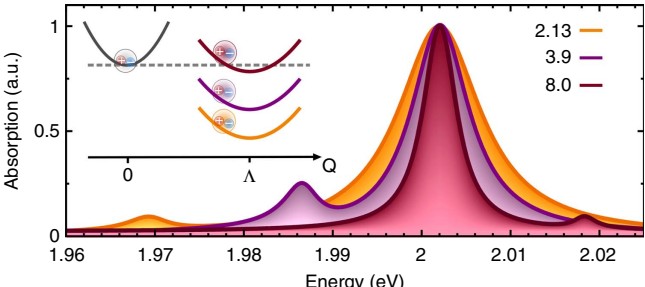

**Figure 3 | Substrate dependence.** Absorption spectra for different substrates are shown including fused silica ($\varepsilon_{bg} = 2.13$), silicon dioxide ($\varepsilon_{bg} = 3.9$) and diamond ($\varepsilon_{bg} = 8.0$) at 77 K. For better comparison, the spectra are shifted relative to the main peak of silicon dioxide. Remarkably, the energetic position of the dark exciton peak can be controlled by the choice of the substrate, since the substrate-induced screening has a different impact on bright KK and dark KΛ excitons.

We assume that the molecules are attached non-covalently to the TMD surface, implying that the electronic wave function of the TMD remains unchanged to a large extent after the functionalization[23]. Moreover, we assume that the adsorption process of the molecules is not influenced by the excitation pulse in the linear optical regime. These assumptions are in agreement with recent studies regarding two-dimensional (2D) heterostructures and molecule-functionalized TMDs[24–27]. We treat the interaction between the attached molecules and the TMD surface as a exciton–dipole interaction, since the molecules induce a static dipole field[28,29]. Here, we focus on the impact of the exciton–molecule interaction on optical properties of the TMD and neglect the changes in optical and vibrational properties of the molecules. We expect these effects to have a minor influence on the activation of dark states in TMDs, where the molecular dipole moment and the coverage play the crucial role.

**Excitonic absorption spectra.** Solving the Wannier equation and the Bloch equations, we obtain microscopic access to optical properties of TMDs as well as their change through functionalization with molecules. Figure 1c illustrates the excitonic absorption spectrum for pristine (orange) and merocyanine-functionalized (purple) WS$_2$ at 77 K and for the molecular coverage of $0.8 \text{ nm}^{-2}$. As expected, the spectrum is dominated by a pronounced peak that can be ascribed to the excitation of the bright KK exciton. However, surprisingly we also find a clearly visible additional peak located $\sim 20$ meV below the main peak. Its position corresponds to the energy of the actually dark KΛ exciton. The exciton–dipole interaction is the driving mechanism that makes the dark states bright. The molecule-induced coupling of the bright KK and the dark KΛ exciton also has an influence on the KK excitons, leading to a small blue shift and a slightly reduced intensity of the main resonance. The exciton–molecule coupling has been studied both theoretically and experimentally in carbon-based 2D structures[28,30]. However, neither graphene nor carbon nanotubes showed the appearance of an additional peak in the absorption spectrum. Note that most existing sensors rely on small energy shifts and intensity changes of optically active transitions. In contrast, we propose the appearance of an additional, well separated peak that presents an unambiguous optical fingerprint of attached molecules suggesting highly efficient dark-state-based sensors.

Now, we study the dependence of this sensor mechanism on externally accessible parameters, such as temperature, substrate, molecular dipole moment and dipole density (molecular coverage). If not otherwise stated we have used for the calculations

WS$_2$ on a silicon dioxide substrate at 77 K with attached merocyanine molecules with a dipole moment of 13 Debye and a molecular coverage of $0.8 \text{ nm}^{-2}$.

For the temperature study, we have explicitly taken into account the increasing excitonic linewidths due to the enhanced efficiency of exciton–phonon scattering. The linewidth values are adjusted to a recent joint experiment-theory study[14]. The increase in the excitonic linewidth at higher temperatures is due to the enhanced exciton–phonon coupling that has been implicitly included in the temperature-dependent dephasing $\gamma$ of the microscopic polarization. Figure 2 illustrates the normalized absorption spectra for 4, 77, 150 and 300 K. With increasing temperature, we observe a clear increase of the linewidth of the main bright exciton resonance. As a result, the dark exciton peak is well pronounced for temperatures up to $\sim 150$ K. At room temperature, the overlap with the main peak is so large that the dark exciton is only visible as a low-energy shoulder. This effect might be further optimized through the control of underlying elementary processes, such as many-particle scattering channels determining the excitonic linewidth of bright and dark states as well as the impact of doping on the relative position of dark and bright states.

For the exciton–molecule interaction, the choice of the substrate plays an important role. The substrate is considered in our approach by the dielectric screening that is implemented within the Keldysh potential for the Coulomb interaction[31]. Figure 3 shows the absorption spectra of merocyanine-functionalized WS$_2$ for different substrates characterized by the dielectric constant $\varepsilon_{bg}$. We observe that the position of the dark exciton resonance crucially depends on the substrate, since the dielectric screening has a different impact on different excitonic eigenvalues in the Wannier equation. This implies a shift in the relative spectral position of the bright KK and the dark KΛ exciton (see the inset in Fig. 3). We find that in the presence of a substrate the dark state is screened stronger due to its higher effective mass, resulting in a more significant change in its excitonic binding energy. Previous studies on dielectric effects in MoS$_2$ revealed a similar behaviour[6,32]. As a result, we can control the position of the additional dark exciton peak in optical spectra by choosing different substrates. For $\varepsilon_{bg} > 8$, we can even move the dark peak above the energy of the main bright excitonic resonance. Note that the linewidth decreases for substrates with enhanced dielectric constant due to the change in the excitonic binding energy and excitonic wave functions that influences both radiative and nonradiative contributions to the linewidth[14].

**Dependence on molecule characteristics**. Finally, we study the impact of molecular characteristics on the proposed sensor mechanism. The investigated molecules are mainly characterized by their dipole momentum and dipole density, that is, molecular coverage on the TMD layer. The dipole orientation turns out to only have a minor effect on the absorption spectra. The dumbbell-like shape of the dipole potential gives rise to a maximal overlap with the TMD surface for the perpendicular dipole orientation[28,30]. Figure 4 shows the general dependence of the absorption spectra on these two quantities. We find that the peak splitting of the bright KK and the dark KΛ exciton as well as the intensity of the dark exciton peak become much stronger for molecules with a larger dipole moment (see Fig. 4a). Our calculations reveal a minimal value for the dipole moment of ~10 Debye for the observation of a well-pronounced additional peak for $WS_2$ on a $SiO_2$ substrate. Note that the visibility of dark states in the absorption spectra is mostly restricted by the excitonic linewidth. Narrow peaks can be obtained by lower temperatures and/or higher dielectric screening, allowing to detect molecules with smaller dipole moments. Furthermore, we observe an interesting behaviour depending on the molecular coverage: the spectral position of the dark exciton peak moves from the higher to the lower energy side of the bright exciton peak with increasing molecular coverage. At the resonance, a clear avoided crossing can be seen (see Fig. 4b).

To explain this behaviour, we consider a limiting case, where we can obtain an analytic expression for the spectral resonances of the bright and the dark exciton in dependence of the molecular dipole moment and coverage. We reduce the complexity of Bloch equations (equations (4) and (5) in the Methods section) by evaluating the Kronecker delta $\delta_{|\mathbf{k}|,\frac{2\pi m}{\Delta R}}$ appearing in the exciton–molecule coupling element. This allows us to perform the sum over $\mathbf{k}$ in the Bloch equations. Note that here the molecular coverage $n = \Delta R^{-2}$ plays an important role. Focusing on the dominant term in the sum with $m = 1$, the equations can be solved analytically via Fourier transformation. We find for $\mathbf{Q} = 0$

$$p_0^K(\omega) = \frac{\Omega(\omega)}{\hbar\omega - \varepsilon_K - \frac{G_{0\tilde{n}}^{\Lambda K} G_{0\tilde{n}}^{K\Lambda}}{\hbar\omega - \tilde{\varepsilon}_\Lambda}} \quad (1)$$

with the abbreviations $\tilde{\varepsilon}_\Lambda = \varepsilon_\Lambda + \frac{\hbar^2(\Lambda - \tilde{n})^2}{2M_\Lambda}$ and $\tilde{n} = 2\pi\sqrt{n}$. We observe that for vanishing exciton–dipole coupling, that is, $G_{0\tilde{n}}^{\Lambda K} G_{0\tilde{n}}^{K\Lambda} = 0$, the pristine resonance is reproduced. Otherwise, resonances in the optical spectrum are located at energies, where the denominator of the equation is zero yielding

$$E_{1,2} = \frac{\tilde{\varepsilon}_\Lambda + \varepsilon_K}{2} \pm \sqrt{\left(\frac{\tilde{\varepsilon}_\Lambda - \varepsilon_K}{2}\right)^2 + G_{0\tilde{n}}^{\Lambda K} G_{0\tilde{n}}^{K\Lambda}}. \quad (2)$$

The equation shows that the peak position in general depends on both the exciton–dipole coupling element $G_{0\tilde{n}}^{\Lambda K} G_{0\tilde{n}}^{K\Lambda}$ and the dispersion of the involved excitons at the K and Λ point.

Taking into account that the coupling element shows a direct dependence on the dipole moment, the analytic approach can explain the behaviour observed in Fig. 4a: a stronger molecular dipole momentum induces a larger dipole field resulting in a more pronounced exciton–molecule coupling and therefore in a greater peak splitting of the resonances in the absorption spectra. Moreover, the analytic solution explains the avoided crossing as a function of the molecular density in Fig. 4b: for $\tilde{\varepsilon}_\Lambda > \varepsilon_K$ ($\tilde{\varepsilon}_\Lambda < \varepsilon_K$), the dark KΛ exciton peak is located energetically higher (lower) than the bright KK exciton. The resonant case is reached by $\tilde{\varepsilon}_\Lambda = \varepsilon_K$ resulting in bright and dark exciton resonances that are only separated by the coupling strength $G_{0\tilde{n}}^{\Lambda K} G_{0\tilde{n}}^{K\Lambda}$. This is the origin of the observed avoided crossings.

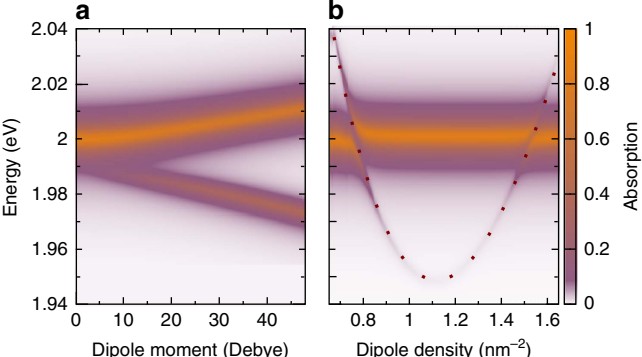

**Figure 4 | Dependence on molecule characteristics.** The spectral position of excitonic resonance can be controlled by the molecular (**a**) dipole moment and (**b**) dipole density (molecular coverage). The stronger the dipole moment, the more pronounced is the peak splitting and the intensity of the dark exciton peak. We reveal that the position of the dark exciton peak moves from the higher to the lower energy side of the bright exciton with increasing molecular coverage. At the resonance condition, avoided crossing can be observed. The appearing parabolic behaviour reflects the dispersion of the dark exciton. The dotted line shows the analytic solution from equation (2).

Furthermore, Fig. 4b also demonstrates a clear parabola with a minimum at $n \approx 1.1\,\text{nm}^{-2}$ corresponding to the coordinates of the Λ point in momentum space. Assuming a small coupling strength with $G_{0\tilde{n}}^{\Lambda K} G_{0\tilde{n}}^{K\Lambda} \ll \left(\frac{\tilde{\varepsilon}_\Lambda - \varepsilon_K}{2}\right)^2$ (which is a good approximation except close to the points of the avoided crossing), we indeed find for the dark exciton resonance the parabolic expression $E_\Lambda(n) = \varepsilon_\Lambda + \frac{\hbar^2(\Lambda - \tilde{n})^2}{2M_\Lambda}$ resembling the dark exciton dispersion relation. The parabola in Fig. 4b can be reproduced very well by our analytic solution (see the dotted line). As a result, one could exploit this parabolic dependence on the molecular coverage to measure the dispersion of dark excitons including their effective mass.

## Discussion

Having understood the impact of attached molecules on the exemplary $WS_2$ material, we now discuss more generally the applicability of different TMD materials in sensor technology. Figure 1b illustrates the excitonic dispersion of $WS_2$ (solid lines) and $MoSe_2$ (dashed lines) as a representative for tungsten- and molybdenum-based TMDs. There is a crucial difference between these TMD families regarding the relative energetic position between the Λ and the K valley: while in tungsten-based TMDs the Λ valley lies energetically below the K valley, the situation is opposite in molybdenum-based TMDs. According to equation (2), the resonant case $\tilde{\varepsilon}_\Lambda = \varepsilon_K$ cannot be reached for the latter and thus avoided crossing does not appear as a function of molecular coverage. A further consequence is that the dark and bright excitons are highly off-resonant ($>100\,\text{meV}$), giving rise to a weak exciton–molecule interaction. Therefore, tungsten-based TMDs, such as $WS_2$ and $WSe_2$, are more suitable materials for sensor applications.

In summary, we have presented a first microscopic study revealing the proof of principle for atomically thin molecular sensors based on dark excitons in 2D materials. We predict the appearance of an additional peak in excitonic absorption spectra that we can ascribe to dark excitons. Efficient coupling between tightly bound excitonic states in the 2D material and the dipole field of attached molecules makes dark excitons visible in optical spectra. In contrast to small peak shifts and intensity changes, this pronounced effect presents a clear unambiguous optical

fingerprint of attached molecules, suggesting high potential for application in future sensor technology devices. Besides the idea of how to activate dark exciton states with molecules, our work also presents a recipe of how to directly measure the dispersion relation of dark excitons by exploiting the remarkable dependence of optical spectra on the molecular coverage.

## Methods

**Microscopic theory on optical properties of TMDs.** To model the optical absorption of functionalized TMDs, we develop a theoretical model describing the excitonic microscopic polarization $p_Q^\mu(t)$ in excitonic basis. First, we solve the Wannier equation[20–22]

$$\frac{\hbar^2 q^2}{2m_\mu}\varphi_q^\mu - \left(1 - f_q^e - f_q^h\right)\sum_k V_{exc}(k)\varphi_{q-k}^\mu = \varepsilon_\mu \varphi_q^\mu \quad (3)$$

corresponding to the excitonic eigenvalue equation offering access to excitonic eigenfunctions $\varphi_q^\mu$ and eigenenergies $\varepsilon_\mu$. Here, we have introduced the reduced mass $m_\mu = (m_h + m_e)/(m_h m_e)$, the electron–hole contribution of the Coulomb interaction $V_{exc}(k)$ including the Fourier transformed Keldysh potential[6,31,33], and the electron (hole) occupations $f_q^{e(h)}$.

To obtain the temporal evolution of the excitonic microscopic polarization $p_Q^\mu(t)$, we solve the Heisenberg equation of motion $i\hbar \dot{p}_Q^\mu(t) = \left[H, p_Q^\mu(t)\right]$ (refs 17,20). This requires the knowledge of the many-particle Hamilton operator that in our case reads $H = H_0 + H_{c-1} + H_{c-c} + H_{c-m}$. It includes the free carrier contribution $H_0$, the carrier–light interaction $H_{c-1}$, the carrier–carrier interaction $H_{c-c}$ and the carrier–molecule interaction $H_{c-m}$.

Exploiting the fundamental commutator relations for fermions[20], we obtain the Bloch equations for the excitonic microscopic polarization $p_Q^\mu$ with the index $\mu = (K, \Lambda)$ denoting the KK and K$\Lambda$ exciton:

$$\dot{p}_Q^K = \Delta\varepsilon_Q^K p_Q^K + \Omega\delta_{Q,0} + \sum_{\mu,k} G_{Qk}^{K\mu} p_{Q-k}^\mu, \quad (4)$$

$$\dot{p}_Q^\Lambda = \Delta\varepsilon_Q^\Lambda p_Q^\Lambda + \sum_{\mu,k} G_{Qk}^{\Lambda\mu} p_{Q-k}^\mu, \quad (5)$$

with the abbreviation $\Delta\varepsilon_Q^\mu = \frac{1}{i\hbar}\left(\varepsilon_\mu + \frac{\hbar^2 Q^2}{2M_\mu} - i\gamma_\mu\right)$. The exciton–phonon interaction is implicitly taken into account by a temperature ($T$) and substrate ($\varepsilon_{bg}$)-dependent dephasing $\gamma = \gamma(T, \varepsilon_{bg})$ of the microscopic polarization determining the linewidth of excitonic resonances. Since the dark excitons can not decay radiatively, we assume $\gamma_\Lambda \approx \gamma_K - \gamma_{rad}$. Especially at low temperatures, this implies a significantly longer lifetime of dark excitons. The optical excitation is expressed by the Rabi frequency $\Omega(t) = \frac{e_0}{m_0}\sum_q \varphi_q^{\mu*} M_q^{cv} \cdot A(t)$ including the optical matrix element $M_q^{cv}$, the external vector potential $A(t)$ as well as the electron charge $e_0$ and mass $m_0$. The dark K$\Lambda$ exciton is only indirectly driven by the exciton–molecule interaction[28]

$$G_{Qk}^{\mu\nu} = \sum_q \left(\varphi_q^{\mu*} g_{q_\alpha, q_\alpha+k}^{cc} \varphi_{q+\beta k}^\nu - \varphi_q^{\mu*} g_{q_\beta-k, q_\beta}^{\nu\nu} \varphi_{q-\alpha k}^\nu\right) \quad (6)$$

with $q_\alpha = q - \alpha Q$ and $q_\beta = q + \beta Q$.

**Exciton–molecule coupling element.** The coupling depends on the overlap of the involved excitonic wave functions and the molecule characteristics entering via the exciton–molecule coupling element $g_{qq'}^{\lambda\lambda'}$. To calculate the coupling elements, we apply the nearest-neighbour tight-binding approach[20,21,34] including fixed (not adjustable) parameters from density functional theory calculations[35]. The matrix element is given by the expectation value of the dipole potential $\phi_l^d(r)$ formed by all attached molecules and reads in electron–hole picture $g_{kk'}^\lambda = \langle\Psi_k^\lambda(r)|\sum_l \phi_l^d(r)|\Psi_{k'}^\lambda(r)\rangle$. Using the tight-binding approach for the electronic wave function $\Psi_k^\lambda(r)$[6], transforming the dipole potential into Fourier space, and assuming periodically distributed molecules we find for the exciton–molecule coupling element $g_{l_1 l_2}^{\lambda_1 \lambda_2} = \frac{e_0}{2\pi\varepsilon_0 \hbar}\sum_m n\delta_{|l_1-l_2|, \frac{2\pi m}{\Delta R}}\sum_j C_j^{\lambda_1*}(l_1) C_j^{\lambda_2}(l_2)\delta_{l_1-l_2,q}\times\int dq \frac{d\cdot q}{|q|^2}e^{-R_z q_z}$. Here, $n$ is the molecular coverage, $d$ the dipole vector, $\Delta R$ the molecular lattice constant, $R_z$ the distance between the TMD layer and the layer of attached molecules and $\varepsilon_0$ the dielectric constant. The tight-binding coefficients $C_j$ with $j = W, S$ (in the case of $WS_2$) determine the contribution from different orbital functions[6]. Further details on the exciton–molecule interaction can be found in ref. 30. Note that these coefficients also include a non-zero geometric phase[36–38]. However, due to the symmetry of the exciton–molecule coupling, this phase cancels out for our studies.

We assume that the external molecules are ordered in a periodical lattice. The lattice constant $\Delta R$ determines the number of molecules within a fixed area. In momentum space, it gives the centre-of-mass momenta induced by the external molecules (see the Kronecker delta in the above equation). In this work, we focus on molecule-induced transitions to the energetically lower lying K$\Lambda$ excitons, since here we find the strongest exciton–molecule interaction compared with other dark

excitons (KK, KK'). As a result, the exciton–molecule coupling element $G_{Qk}^{K\Lambda}$ is much larger than $G_{Qk}^{KK}$ and $G_{Qk}^{KK'}$. This can be ascribed to the enhanced overlap of the involved excitonic wave functions due to the high density of states in the $\Lambda$ valley.

**Data availability.** The data that support the findings of this study are available from the corresponding author on request.

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

## Acknowledgements

This project has received funding from the European Union's Horizon 2020 research and innovation programme under grant agreement no. 696656 within the Graphene Flagship (E.M. and G.B.). Furthermore, we acknowledge support by the Chalmers Area of Advance in Nanoscience and Nanotechnology (M.F. and E.M.) and by the Deutsche Forschungsgemeinschaft through SFB 658 (E.M.) and 951 (A.K., M.F. and G.B.). Finally, we thank M. Selig and R. Gillen (TU Berlin) for fruitful discussions.

## Author contributions

M.F. performed the theoretical calculations. All authors contributed to the interpretation of the results and writing of the manuscript.

## Additional information

**Competing interests:** The authors have filed a patent on the finding in this work (patent number 16194861.7-1554).

**Publisher's note**: 

