## [Peer review file · Nature Communications]

Reviewers' comments:

Reviewer #1 (Remarks to the Author):

The manuscript submitted by Feierabend and co-authors reports an interesting and comprehensive theoretical investigation on the possible use of a single W-based Transition Metal Dichalcogenide (TMDC) Monolayer (ML) as an efficient chemical sensor. The basic idea is that by activating dark exciton states by means of strong interaction between molecules that possess a large dipole moment and the TMDC ML, the appearance of additional peaks in the absorption spectra is clearly demonstrated. This manuscript presents new informations as well as new hypotheses of work on the study of the highly complex nature of exciton states in TMDC MLs, so far unresolved. I have found the present communication interesting for a broad audience, and the data convincing. However I would like to invite the authors to revise their manuscript to address specific concerns, listed below, before a final decision is reached. Here are the main issues:

— the use of terms like « forbidden dark excitons » is misleading to my opinion. By forbidden do the authors means not allowed by one-photon excitation ? Would not be sufficient to speak about dark states only ?

— More importantly, the main assumption of the present study is exposed in the second §. The authors propose that indirect K-Lambda excitons lie below direct K-K states for W-based systems only. To support this they provide three references,(12-14) but only the latter propose indeed this energy ordering. In ref. [12] only indirect K-K' transition were studied, when in the experimental work of ref. [13], it was only hypothesized the presence of dark states below the bright ones in WSe₂ ML, without any details in their nature. Since the present study is based entirely on this hypothesis, that refers only to a unpublished, (but probably submitted) paper, I would like to see convincing arguments in the presence of such a state below the bright state for WS₂. What would be the origin of this energy ordering ? A drastic change in the effective mass, in conjunction with a small energy separation between the two minima (in K and in Lambda) in the conduction bands, as it is reported in PRB 90, 045422 ?

But in this case why for MoSe₂ ML for which the energy separation between the two minima is also modest, why the dark state is 100 meV higher ?

Additional point, in PRL 115, 257403 for WSe₂ ML it is proposed that there is a splitting of 30 meV between dark and bright excitons. How does the model use in the present work agree with this experimental fact ?

— Could the authors provide details on the fact that only K-Lambda transitions are becoming optically active by using molecular dipole moments ? Why not K-K' or K-K dark states ?

— It is proposed here to use non-covalent doping, as in CNT or graphene for instance, to keep the electronic structure untouched upon doping. However TMDCs are much more reactive compounds than carbon based nanostructures. What is the state of the art on this side for TMDCs ?

— In Figure 1, what is the physical meaning of the blueshift observed upon the interaction with the molecules ?

— When the substrate effect is tested, it is proposed that the main effect of the change in the dielectric background constant is on the exciton effective masses. Could the authors report in a Table the obtained values and some comparison elements with previous studies for instance ?

— Again in this part, how interpret the drastic changes in the oscillator strengths (OS) of the activated dark states ?

Why the OS is larger for the silicon dioxide ?

— In figure 4, at which temperature the calculations are performed ?

— Could the authors comment on the minimal values of dipole moment hold by the molecule required to obtain a significant activation of the dark state, through coupling ? What kind of applications would it be allowed, then ?

and some minor comments:

— in the methods part, the definition of the coupling term between the molecules and the TMDC remains obscure. For instance how the C_j are obtained ?

--minor typos in ref [13] Arora A. is the first author and in ref. [15] the page is not provided.

Reviewer #2 (Remarks to the Author):

This is an interesting idea, at least conceptually. If this works as suggested, one should be able to see the dark exciton “coming to life” with a simple PL measurement. In principle the experiment could be quite simple: physisorb something on a TMD and one should see it. However, I have doubts that this would be as easy as it is proposed. I think the manuscript could be suitable for publication although the following concerns would need to be addressed first:

1) the methods need to be validated. there is no discussion of the accuracy and validity of the approach taken, and no comparison with experiments to justify the models, whether at the DFT level or in the Wannier and Bloch solutions. Much more details and discussion on accuracy must be provided.

2) since the idea of this paper is directed towards sensing, as stated even in the title, the authors must compare this idea with other nanoscale sensing techniques. There is no discussion or comparison or context whatsoever (apart from the general need for good sensor in the first paragraph). Why would the use of a dark exciton in this way in a TMD be advantageous over other sensing techniques, e.g., nanomechanical sensors or other single-molecule detection approaches (either fluorescent or non-fluorescent based)? Is the advantage in sensitivity, selectivity, cost, or other? Even though this is a theoretical paper, if the authors want to claim they have a new sensor, they must contextualize the claim carefully in the field of sensing.

3) The bright exciton has plenty of "material" to allow a simple measurement to show it, since one has the integral of each single exciton generated across the surface. In this case, one has only as many excitons as absorbates. If it's physisorbed, would the energy of the beam be sufficient to desorb it? Can the mediation be sustained with external energy pumped in? Can the authors explain the high sensitivity to temperature? These questions need to be discussed and much more discussion in general is needed for how such a sensor would actually operate.

4) There is no mention of the lifetime of the mediated exciton. This would be important to know in order to understand the system better and predict possible collective effects, etc.

Reviewer #3 (Remarks to the Author):

The manuscript by Feierabend et al., proposes to use atomically thin transition metal dichalcogenides (TMDs) as sensors to sense molecules. The scheme for sensing is based on "brightening" of finite center-of-mass momentum exciton which is optically forbidden or dark due to conservation of linear momentum. The mechanism of brightening due to the presence of the molecule which possesses a dipole moment is attributed to exciton-dipole coupling.

The authors solve the Wannier equation (under effective mass approximation) to obtain exciton energy and eigenstates and then use so-called semiconductor Bloch equations to obtain the splitting of the direct $K\Gamma$ exciton and the indirect (which is optically dark to begin with) $K\Lambda$ exciton and their respective intensity as a function of the dipole strength of the molecule and the dipole density. Indeed, it found that the indirect exciton brightens and splits from the direct bright exciton.

Although the the paper is technically sound, it fails to provide any new insight into the field of TMDs, which is an actively researched field. Most of their conclusions are based on the analysis mentioned in the Methods section, which is hardly transparent or insightful. Due the reasons outlined below, I am of the opinion that their work is not suitable for publication in Nature Communications.

The mechanism of brightening of indirect exciton can be simply explained by the following picture - The molecule is modeled as a dipole which is placed close to a free exciton. Due to the dipolar potential, the exciton is trapped and localized in real space to a distance, say ΔR . This localization in real space relaxes the conservation of linear momentum by $\Delta k \sim 2\pi/\Delta R$. As a result if Δk is comparable to the center-of-mass momentum of the $K\Lambda$ exciton, then it is expected to brighten. The details of the dipole moment and excitonic dispersion quantitatively determine the intensity and splitting of the dark and bright excitons. In fact, the effective coupling between the dark and bright exciton due the presence of the dipole can also be estimated to a good approximation without complicated equations. This is a general effect which is expected for any exciton, the only thing pertaining to TMD here is the presence of $K\Lambda$ exciton. It is for this reason that I feel that their theoretical analysis doesn't provide any new insight into the field of TMDs.

As for their proposal for using this effect for sensing molecule, it is also not entirely clear how important this effect will be in a realistic experiment. According to the authors, "The main goal of our study is to investigate the question whether dark excitons can be exploited for sensing of molecules". However, the authors fail to provide any estimate of sensitivity of their scheme. Can this scheme be ever used to detect a single molecule of reasonable dipole moment?

More importantly, I feel that the authors have overlooked certain mechanisms that can seriously undermine their scheme. For example, the relaxation of the exciton through non-radiative channels which are likely to open when the exciton brought in the vicinity of the molecule which has other degrees of freedom (vibrational, etc) and can not be simply modeled as a dipole.

Other specific comments/criticisms -

1. The title "Out of the dark and into the light: ..." is probably suitable for a popular commentary but not for a regular scientific article.
2. The figures with gradient shading although look artistic, they don't convey any information and can unnecessary confuse the reader who is trying to gain a quantitative understanding.
3. The use of effective mass Wannier equation is not strictly valid for TMD excitons as was recently shown in two publications - Phys. Rev. Lett. 115, 166802 and 166803 (2015). Although for the current analysis, Berry curvature effects are likely to play a minor role, the shortcomings of the Wannier equation for TMDs should be mentioned.
4. In Fig. 4, what dipole density was assumed for the left panel and what dipole moment was assumed for the right panel?
5. Can the authors offer a simple explanation between the dark and bright exciton due to the presence of the dipole i.e., demystify $G_{0n}^{\Lambda K} G_{0n}^{K\Lambda}$ (see above)?

Reviewer#1

“This manuscript presents new informations as well as new hypotheses of work on the study of the highly complex nature of exciton states in TMDC MLs, so far unresolved. I have found the present communication interesting for a broad audience, and the data convincing.”

We thank the referee for the careful evaluation of our manuscript and for acknowledging that it is novel, convincing, and of interest for a broad audience.

Comment 1

The use of terms like « forbidden dark excitons » is misleading to my opinion. By forbidden do the authors means not allowed by one-photon excitation ? Would not be sufficient to speak about dark states only ?

Response

Indeed, by optically forbidden we mean not allowed by one-photon excitation and therefore it is sufficient to talk about dark excitons.

Revision

We have changed the manuscript accordingly.

Comment 2

The authors propose that indirect K-Lambda excitons lie below direct K-K states for W-based systems only. To support this they provide three references,(12-14) but only the latter propose indeed this energy ordering. Since the present study is based entirely on this hypothesis, that refers only to a unpublished, (but probably submitted) paper, I would like to see convincing arguments in the presence of such a state below the bright state for WS2. What would be the origin of this energy ordering ? A drastic change in the effective mass, in conjunction with a small energy separation between the two minima (in K and in Lambda) in the conduction bands, as it is reported in PRB 90, 045422 ?But in this case why for MoSe2 ML for which the energy separation between the two minima is also modest, why the dark state is 100 meV higher ?Additional point, in PRL 115, 257403 for WSe2 ML it is proposed that there is a splitting of 30 meV between dark and bright excitons. How does the model use in the present work agree with this experimental fact?

Response

We thank the referee for bringing up the important question of the origin of the energy ordering. There are two effects leading to the pronounced ordering:

First, the energy separation in the conduction band at the K and L valley is important. Note that the spin degeneracy in the conduction band is different for tungsten and molybdenum based TMDs (e.g. Ref. 34 in the manuscript). The work mentioned by the referee (PRB 90, 045422) did not consider the effect of spin-orbit coupling, which is why they obtain similar energy separations for all TMDs. The included Fig. 1R illustrates the dispersion relation for both tungsten- and molybdenum-based TMDs, where the blue lines in Fig. (a) and (b) indicate the energetically lowest transitions. The second effect stems, as assumed by the referee, from different effective masses in K and L valley, which affects the binding energy of the corresponding excitons.

Taking both effects into account, we find by solving the Wannier equation that the dark KL exciton is energetically below the bright KK exciton in tungsten-based TMDs, while the order is opposite in molybdenum-based TMDs. The results in PRL 115, 257403 (Ref. 12) are in excellent agreement with our findings. They ascribe the lower energy peak in WSe2 to dark states and experimentally support the theoretical prediction that the origin of the energetic ordering stems from the spin-degeneracy in the conduction band. The order of magnitude of the observed dark-bright splitting (30 meV) is also in good agreement with our theory.

Figure 1R Band structure obtained from k - p theory for two-dimensional transition metal dichalcogenide semiconductors. Taken from Andor Kormányos et al 2015 *2D Mater.* 2 022001.

Revision

We have changed the main text to clarify the origin of the energetic ordering by including the following paragraph: “Recent theoretical and experimental studies [12-14] have demonstrated that there are dark excitons located energetically below the bright KK exciton in tungsten-based TMDs, cf. Fig. 1(b). This can be ascribed to two effects: (i) different spin degeneracy in the conduction band for tungsten- and molybdenum-based TMDs and (ii) different excitonic binding energy for K and L valley due to their specific effective mass.”

Comment 3

Could the authors provide details on the fact that only K-Lambda transitions are becoming optically active by using molecular dipole moments? Why not K-K' or K-K dark states?

Reply

The referee raises another important point. The exciton-molecule interaction is characterized by both the excitonic properties of the TMD and the characteristics of the molecule. For the molecule, not only the dipole moment is important but also the molecular coverage. In fact, the distance of molecules (R) in real space determines the possible excitonic center of mass momentum (Q) via the relation $R=2\pi/Q$. In analogy, one can express the molecular coverage as $n=R^{-2}/\text{propto } Q^2$. Hence, the molecular coverage can be used to address KK, KL, or KK' dark states. We focused on the KL excitons in our work since our calculation revealed that it has the strongest exciton-molecule interaction due to the enhanced density of states in the L valley enhancing the overlap of the involved excitonic wave functions. Moreover, interestingly the dark KL exciton is energetically below the bright KK exciton and therefore shows the unique dependence on the molecular coverage shown in Fig. 4 in the manuscript.

Revision

We have added a paragraph clarifying this point: “In this work, we focus on molecule-induced transitions to the energetically lower lying KL excitons, since here we find the strongest exciton-molecule interaction compared to other dark excitons (KK, KK'). This can be ascribed to the enhanced overlap of the involved excitonic wave functions due to the high density of states in the L valley.”

Comment 4

It is proposed here to use non-covalent doping, as in CNT or graphene for instance, to keep the electronic structure untouched upon doping. However TMDCs are much more reactive compounds than carbon based nanostructures. What is the state of the art on this side for TMDCs ?

Reply

Our manuscript is motivated by recent experimental studies regarding 2D hetero structures which bond non-covalently, for example MoS₂-Black Phosphorus (*ACS Nano*, **8 (8)**, pp 8292–8299 (2014)) or MoS₂-WS₂ (*Nature Nanotechnology* **9**, 682–686 (2014)). Those studies reveal that due to the weak interaction between the layers the system remains most of its electronic properties. Moreover, theoretical first principle studies of such heterostructures (PRB 88, 085318) show that the electronic structure remains unchanged to a large extent.

Since TMDs are reactive compounds, as mentioned by the referee, previous studies on functionalizing TMDs with molecules mainly focused on covalent doping aiming at a change in band structure to tailor the electronic properties (Review by Ryder et al. "Chemically Tailoring Semiconducting Two-Dimensional Transition Metal Dichalcogenides and Black Phosphorus." *ACS nano* 10.4 (2016): 3900-3917). However, there are also first studies on non-covalent functionalization of TMDs with molecules via van der Waals interaction. Only very recently, Nguyen et al. demonstrated successfully van der Waals functionalization with molecules in MoS₂ ("Excitation dependent bidirectional electron transfer in phthalocyanine-functionalised MoS₂ nanosheets." *Nanoscale* 8.36 (2016): 16276-16283). Moreover, Chen, Xin, et al. ("Functionalization of Two-Dimensional MoS₂: On the Reaction Between MoS₂ and Organic Thiols." *Angewandte Chemie International Edition* 55.19 (2016): 5803-5808) revealed non-covalent bonding characteristics in MoS₂ and organic molecules.

Revision

We have added a paragraph in the manuscript addressing this point: "We assume that the molecules are attached non-covalently to the TMD surface implying that the electronic wave function of the TMD remains unchanged to a large extent after the functionalization [22]. Moreover, we assume that the adsorption process of the molecules is not influenced by the excitation pulse in the linear optical regime. These assumptions are in agreement with recent studies regarding 2D heterostructures and molecule-functionalized TMDs [citations from studies mentioned above]."

Comment 5

In Figure 1, what is the physical meaning of the blueshift observed upon the interaction with the molecules?

Reply

The exciton-molecule interaction introduces a coupling between the bright KK and the dark KL exciton. The coupling between two excitonic states can be understood in analogy to the problem of two coupled pendulums. The effect of the coupling on the KL exciton is more obvious, because it becomes visible in the absorption spectrum (additional peak in Figure 1). However, the KK exciton is also affected by the coupling leading to the observed blue shift. To be precise one should say that the interaction with the molecules leads to a peak splitting, which is analytically expressed by Eq. (2) in the manuscript. Depending on which side of the pristine resonance the additional peak appears, the pristine peak is either blue- or red-shifted.

Revision

We have changed the manuscript by adding: "The molecule-induced coupling of the bright KK and the dark KL exciton has also an influence on the KK excitons leading to a small blue shift and a slightly reduced intensity of the main resonance."

Comment 6

When the substrate effect is tested, it is proposed that the main effect of the change in the dielectric background constant is on the exciton effective masses. Could the authors report in a Table the obtained values and some comparison elements with previous studies for instance ?

Reply

The excitonic binding energy is mainly determined by the effective mass and the strength of the Coulomb interaction, cf. Eq. (3) in the manuscript. The dielectric background constant enters here via the screening of the Coulomb potential. We find in agreement with Lin, Yuxuan, *et al.* ("Dielectric screening of excitons and trions in single-layer MoS₂." Nano letters **14.10** (2014): 5569-5576) and Berghauser *et al.* (Ref. 6) that higher dielectric constants lead to lower bound excitons. A direct comparison is not possible because those studies only list values for MoS₂. Table 1 shows the excitonic binding energies for both K and L exciton in WS₂. Due to different effective masses, K and L excitons have different binding energies for the same substrate. Changing the substrate induces approx. the same relative change in the binding energy for K and L excitons, however the absolute change is larger for the L exciton due to its higher unscreened value.

substrate	$E_{\text{exc}}^{\text{K}}$ [meV]	$E_{\text{exc}}^{\text{L}}$ [meV]	$E_{\text{exc}}^{\text{L}} - E_{\text{exc}}^{\text{K}}$ [meV]	$\delta E^{\text{L}} - \delta E^{\text{K}}$
1.0	745	858	113	0
2.13	539	637	98	0.02
3.9	363	443	80	0.02
8.0	186	235	49	0.02

with the relative change $\delta E = E_{\text{exc}}(\text{substrate})/E_{\text{exc}}(\text{freestanding})$

Revision

We have changed the manuscript further clarifying the impact of the substrate: "We find that in the presence of a substrate the dark state is screened stronger due to its higher effective mass resulting in a more significant change in its excitonic binding energy. Previous studies on dielectric effects in MoS₂ revealed a similar behaviour [citation from the work mentioned above]"

Comment 7

Again in this part, how interpret the drastic changes in the oscillator strengths (OS) of the activated dark states? Why the OS is larger for the silicon dioxide ?

Reply

This is an important point and we realized that a reference is needed here to avoid confusion. From a very recently published work (Ref 14), we know that the excitonic linewidth decreases with increasing dielectric constant of the substrate. The reason is that the substrate screens the Coulomb interaction leading to a change in the excitonic binding energy and excitonic wave functions, which influence both radiative and non-radiative channels determining the linewidth.

Revision

We have included one paragraph in the methods part clarifying this point: "The exciton-phonon interaction is implicitly taken into account by a temperature (T) and substrate (ϵ) dependent dephasing $\gamma = \gamma(T, \epsilon)$ of the microscopic polarization determining the linewidth of excitonic resonances."

Moreover we have added to the manuscript: "Note that the linewidth decreases for substrates with enhanced dielectric constant due to the change in the excitonic binding energy and excitonic wave functions, which influences both radiative and non-radiative contributions to the linewidth [Ref. 14]"

Comment 8

In figure 4, at which temperature the calculations are performed ?

Reply

The calculations are performed at 77 Kelvin.

Revision

We have added a sentence to the manuscript to clarify the standard parameters: "If not otherwise stated we have used for the calculations WS2 on a silicon dioxide substrate at 77 K with attached merocyanine molecules with a dipole moment of 13 Debye and a molecular coverage of 0.8 nm⁻²."

Comment 9

Could the authors comment on the minimal values of dipole moment hold by the molecule required to obtain a significant activation of the dark state, through coupling ? What kind of applications would it be allowed, then?

Reply

This is again an important point regarding the utilization of the proposed sensor application. Our calculations reveal a significant activation of the dark state for molecules with a minimal value of dipole moment of approximately 10 Debye for WS2 on a silicone dioxide substrate at 77 Kelvin, cf. Fig. 4(a). This could be, for example, merocyanine molecule or any other molecule with a permanent dipole moment in this range. Variation of the substrate and the temperature can be used to further decrease the minimal value of the dipole moment *since*: (i) higher dielectric screening decreases the separation between dark and bright state (cf. comment 6) and hence they become more resonant in the absorption spectrum, i.e. the dark peak shows a higher intensity. Moreover, the linewidth decreases with higher dielectric screening (cf. comment 7) leading to clearer absorption peaks, (ii) lower temperatures decrease the linewidth (cf. Fig. 2) and lead to a more pronounced dark exciton peak. Hence, the visibility of the dark exciton peak can be optimized and therewith the minimal value of dipole moment can be further decreased. While the main goal of our work is the demonstration of the proof-of-principle for the suggested novel sensor mechanism, further optimization of the working principle will be obtained in future studies.

Revision

We have added a paragraph to the manuscript: "Our calculations reveal a minimal value for the dipole moment of approximately 10 Debye for the observation of a well pronounced additional peak for WS2 on a SiO₂ substrate. Note that the visibility of dark states in the absorption spectra is mostly restricted by the excitonic linewidth. Narrow peaks can be obtained by lower temperatures and/or higher dielectric screening, allowing to detect molecules with smaller dipole moments.

Comment 10

In the methods part, the definition of the coupling term between the molecules and the TMDC remains obscure. For instance how the C_j are obtained ?

Reply & Revision

We thank the referee for drawing our attention to this point. We have added clarifications to the methods section: "The matrix element is given by the expectation value of the dipole potential $\langle \Psi_k^\lambda | \hat{V}_d(\mathbf{r}) | \Psi_k^\lambda \rangle$ formed by all attached molecules and reads in electron-hole-picture $g_{\mathbf{k}\mathbf{k}'}^\lambda = \langle \Psi_k^\lambda | \hat{V}_d(\mathbf{r}) | \Psi_{\mathbf{k}'}^\lambda \rangle$. Using the tight-binding approach for the electronic wave function Ψ_k^λ [Ref. 6], transforming the dipole potential into Fourier space, and assuming periodically distributed molecules we find for the exciton-molecule coupling element". Furthermore, we add "The tight-binding coefficients C_j with j=W, S (in the case of WS₂) determine the contribution from

different orbital functions [6]. Further details on the molecule-substrate matrix elements can be found in Ref. 29”

Comment 11

Minor typos in ref [13] Arora A. is the first author and in ref. [15] the page is not provided.

Reply & Revision

We thank the author for careful reading. We have added the corresponding changes in the manuscript.

Reviewer #2

"This is an interesting idea, at least conceptually. If this works as suggested, one should be able to see the dark exciton "coming to life" with a simple PL measurement. I think the manuscript could be suitable for publication."

We thank the referee for the very positive assessment of our work.

Comment 1

The methods need to be validated. there is no discussion of the accuracy and validity of the approach taken, and no comparison with experiments to justify the models, whether at the DFT level or in the Wannier and Bloch solutions. Much more details and discussion on accuracy must be provided.

Reply

We agree with the referee that this important point and needs to be addressed in the manuscript. Below we discuss the validity of our approach with respect to excitonic properties in pristine TMD monolayers as well as functionalized TMDs:

(i) Excitonic properties in pristine TMDs

We use the density matrix formalism combined with a tight-binding approach to derive an equation of motion for the microscopic polarization in different TMDs. This corresponds to the well established semiconductor Bloch equation [Ref. 19: Haug, Koch, Quantum theory of the optical and electronic properties of semiconductors, World Scientific 2005]. The introduced separation in relative and center of mass momenta enables us to find the excitonic eigen energies and wave functions by solving the Wannier equation. This is also a well established method to solve the excitonic eigen value problem [Ref. 20: Kira, M. & Koch, S.W. Many-body correlations and excitonic effects in semiconductor spectroscopy. Progress in Quantum Electronics 30, 155 (2006)]. Our approach applied to TMDs was first introduced in Ref. 6, where the calculated excitonic binding energies and excitonic resonances in different TMD materials have been successfully compared with previous experimental and DFT work. It was shown that the absorption spectrum is strongly dominated by excitons, which show both a large binding energy (approx. 0.5 eV) and extend over multiple unit cells (approx. 1 nm).

Another crucial point for the validity of our approach is the application of the effective mass approximation assuming that the energy dispersion can be described by parabolas. In the vicinity of the K and L point this is a good approximation. However, in the region between non-parabolic effects will be important. As a result, the validity of our theoretical approach is limited to physical processes around the high-symmetry points in the Brillouin zone. Furthermore, for the current study we focus on the exciton-molecule interaction and neglect other possible coupling mechanisms, such as exciton-phonon and exciton-exciton scattering. The latter can be neglected in the considered regime of linear optics, while the first could principally also induce indirect transition to dark excitons, if the energy of optical phonons coincides with the energetic separation between dark and bright excitons.

(ii) Functionalized TMDs

For the interaction between attached molecules and TMD surface, we assume a static dipole field induced by the molecules. This approach was used in recent publications for functionalized graphene (Ref. 29) and functionalized nanotubes (Ref. 27). Moreover, a joint theory–experiment study (E. Malic et al. "Carbon nanotubes as substrates for molecular spiropyran-based switches." *Journal of Physics: Condensed Matter* 24.39 (2012): 394006) showed good qualitative agreement between theoretical predictions and experimentally measured effects. Here, we focus on the impact of the exciton-molecule interaction on the optical properties of the TMD and neglect the changes in the molecular spectrum itself. This approximation is justified since the changes in optical and vibrational properties of the molecule will not significantly change its dipole moment, which presents the most important property for the interaction with TMD excitons.

Revision

We have added a new paragraph to the manuscript discussing the applied approximations and the validity of our approach: “Our approach is consistent with previous studies [6,14], which revealed strongly bound excitons with binding energies of approximately 0.5 eV and excitonic linewidths in the range of tens of meV - in good agreement with recent experimental findings [5,7]. Applying the effective mass approximation, our investigations are limited to processes around the high-symmetry points in the Brillouin zone. Furthermore, we focus on the exciton-molecule interaction in this work and neglect the impact of exciton-exciton and exciton-disorder processes.”

Comment 2

Why would the use of a dark exciton in this way in a TMD be advantageous over other sensing techniques, e.g., nanomechanical sensors or other single-molecule detection approaches (either fluorescent or non-fluorescent based)? Is the advantage in sensitivity, selectivity, cost, or other?

Reply

We thank the referee for the comment and agree that there is a further need to put the idea more into the context of sensors. However, we would like to emphasize that our manuscript is a first microscopic study revealing the proof-of-principle for atomically thin molecular sensors based on dark excitons. Our main aim is to trigger new research efforts in leading experimental groups to investigate this novel sensor mechanism.

One of the main challenges of sensor technologies is transforming physical effects (here additional peaks in the absorption spectra) into electrical signals. The more significant the changes in the absorption spectra, the easier it is to reveal clear signals. First, TMDs as atomically thin nanomaterials show an optimal surface-to-volume ratio exhibiting a remarkable sensitivity to changes in their surrounding. Second, to our best knowledge, most of existing sensing techniques rely either on small energy shifts or changes in peak intensity. Nanomechanical sensors (Nanoscale, 2012, 4, 4925) as well as non-fluorescent based molecule detection (*Nature Photonics* 10, 11–17) both rely on energy shifts as the answer to attached molecules. On the other hand, fluorescent based molecule detection is mainly based on “counting” the molecule signals during the read-out process and therefore relies on intensity changes (*Anal. Chem.*, 2013, 85 (3), pp 1258-1263).

The dark state based sensor proposed in our manuscript has the advantage of the appearance of an additional peak in easy accessible linear absorption spectra in the presence of molecules. This could be used as a simple optical read-out for the present molecules. Moreover, the position and intensity of the peak depend on the coverage of the molecules, which enables a reliable sensor due to the presence of two changing parameters. Another point of advantage is the fast response time, which is only limited by the technical realization, since the signal in the absorption spectra will be there immediately after the molecules are attached to the TMD.

Revision

We have modified the conclusions of the revised manuscript to address this point: “We have presented a first microscopic study revealing the proof-of-principle for atomically thin molecular sensors based on dark excitons in atomically thin 2D materials. We predict the appearance of an additional peak in excitonic absorption spectra that we can ascribe to dark excitons. Efficient coupling between tightly bound excitonic states in the 2D material and the dipole field of attached molecules makes dark excitons visible in optical spectra. In contrast to small peak shifts and intensity changes, this pronounced effect presents a clear unambiguous optical finger print of attached molecules suggesting high potential for application in future sensor technology devices.”

Comment 3

The bright exciton has plenty of "material" to allow a simple measurement to show it, since one has the integral of each single exciton generated across the surface. In this case, one has only as many excitons as absorbates. If it's physisorbed, would the energy of the beam be sufficient to desorb it? Can the mediation be sustained with external energy pumped in?

Reply

Within the unique electronic structure of TMDs there is a variety of excitons, both dark and bright ones, cf. Fig1(a). The bright ones can be accessed directly with one-photon excitation. For the dark ones, there is an additional momentum transfer needed (indirect transitions). In our manuscript, this momentum is induced by the attached molecules. In real space, this means that there is a molecular lattice mismatching the TMD lattice and providing the required center of mass momentum. Hence, one can say that the presence of molecules enables indirect transitions making already present dark excitons visible in absorption spectra, i.e. principally we can indirectly excite as many dark excitons as we can optically induce bright excitons. Regarding the adsorption of molecules, we assume that the molecules attach non-covalently to the surface, i.e. they are physisorbed. The optical excitation with a laser pulse in Fig. 1(a) is only needed to excite the TMD in order to get its absorption spectrum. Since we are in the linear optical regime, the excitation pulse is very weak and it does not influence the adsorption process of the molecules. Previous studies on exciton-molecule coupling in carbon nanotubes (E. Malic et al. "Carbon nanotubes as substrates for molecular spiropyran-based switches." *Journal of Physics: Condensed Matter* 24.39 (2012): 394006.) revealed experimental evidence that the functionalization is independent of the excitation pulse.

Revision

We have added to the manuscript: "We assume that the adsorption process of the molecules is not influenced by the excitation pulse in the linear optical regime."

Comment 4

Can the authors explain the high sensitivity to temperature?

Reply

The high sensitivity to temperature stems from the increasing exciton-phonon coupling at enhanced temperature leading to broader peaks and less pronounced dark excitons. A detailed study on temperature-dependent excitonic linewidths in TMDs has been very recently published, cf. Ref. 14.

Revision

We have added to the manuscript: "The increase in the excitonic linewidth at higher temperatures is due to the enhanced exciton-phonon coupling that has been implicitly included in the temperature-dependent dephasing γ of the microscopic polarization. "

Comment 5

Much more discussion in general is needed for how such a sensor would actually operate.

Reply

The main goal of this work is a first demonstration of the proof-of-principle for atomically thin molecular sensors based on dark excitons. Our aim is to trigger new research efforts in leading experimental groups to investigate and further optimize this novel sensor mechanism. The idea is that the proposed sensor technique has the clear advantage over conventional methods relying on small peak shifts and intensity changes. The appearance of an additional peak in easily accessible linear absorption spectra can be exploited as a simple and unambiguous optical read-out for the presence of molecules.

Revision

We have modified the conclusions of the revised manuscript to address this point according to the revision to the comment 2.

Comment 6

There is no mention of the lifetime of the mediated exciton. This would be important to know in order to understand the system better and predict possible collective effects, etc.

Reply

The referee addresses an important point. The lifetime of bright and dark excitons can be estimated from our calculations. We can directly determine the coherence lifetime of the microscopic polarization. This is expressed by the dephasing rate γ appearing in the Bloch equation in the manuscript. The dephasing is determined by radiative and non-radiative decay channels, the latter in particular driven by exciton-phonon coupling [Ref. 14]. Since dark KL excitons can not decay radiatively, we assume $\gamma_L = \gamma_K - \gamma_{rad}$. Exploiting the relation between the dephasing rate and the coherence lifetime $\tau \cdot \gamma = \hbar$, we find that the coherence lifetime of the dark exciton is significantly larger at low temperatures compared to the bright exciton, cf. table below

T [Kelvin]	tau_L [fs]	tau_K [fs]
4	330	110
77	165	82
250	47	36

Since the coherence lifetime (T2 time) and the exciton lifetime (T1 time) are related [Ref. 20], this result implies for the lifetime of dark excitons that they will live longer due to the lack of radiative processes. To our knowledge there have not been any experimental works yet on lifetimes of dark excitons in TMDs. Nevertheless, studies on dark excitons in carbon nanotubes (PRL 95, 247402 (2005)) or dark excitons in InGaAs quantum wells (*Nature* 418, 754-757 (2002)) reveal longer lifetimes for dark states compared to bright ones.

Revision

We have added a sentence addressing the excitonic lifetimes: “Especially at low temperatures, this implies a significantly longer lifetime of dark excitons.”

Reviewer #3

“Although the paper is technically sound, it fails to provide any new insight into the field of TMDs, which is an actively researched field.”

First of all, we thank the referee for acknowledging that our work is technically sound. Furthermore, we hope to convince him/her that the gained insights are indeed novel. As a matter of fact, we have just filed a patent for the proposed sensor mechanism based on dark excitons.

Comment 1

The mechanism of brightening of indirect exciton can be simply explained by the following picture - The molecule is modeled as a dipole which is placed closed to a free exciton. Due to the dipolar potential, the exciton is trapped and localized in real space to a distance, say ΔR . This localization in real space relaxes the conservation of linear momentum by $\Delta k \sim 2\pi/\Delta R$. As a result if Δk is comparable to the center-of-mass momentum of the $K\Lambda$ exciton, then it is expected to brighten. The details of the dipole moment and excitonic dispersion quantitatively determine the intensity and splitting of the dark and bright excitons. In fact, the effective coupling between the dark and bright exciton due the presence of the dipole can also be estimated to a good approximation without complicated equations.

Reply

We agree with the referee that for a hand-waving understanding of exciton-molecule interaction the picture drawn by the referee is sufficient. However, the effect is strongly sensitive to the relative position of dark and bright states that depends on the strength of the Coulomb interaction as well as on the molecular characteristics including dipole moment and molecular coverage. To check if the simple but novel idea works out in real materials, one has to uncover the relevant elementary processes on the microscopic scale.

Comment 2

This is a general effect which is expected for any exciton, the only thing pertaining to TMD here is the presence of $K\Lambda$ exciton. It is for this reason that I feel that their theoretical analysis doesn't provide any new insight into the field of TMDs.

Reply

It is true that the basic idea is simple, but as the referee will probably agree most good ideas are simple. To best of our knowledge, our work is the first to demonstrate that the presence of molecules enables a coupling between bright and dark states circumventing optical selection rules and making dark states visible in absorption spectra. Furthermore, an important ingredient for this idea to work is a remarkable property of TMDs exhibiting energetically lower lying dark excitons. This is not a tiny aspect, but it is actually crucial. The new insight coming from our work is to combine the idea of activating dark exciton states with molecules and realizing that TMDs offer a unique spectrum of bright and dark excitons. Since recent studies including theory and experiment have shown the importance of dark excitons in TMDs (Refs. 9-14), our work can also contribute to an experimental measurement of the dark KL exciton dispersion by exploiting the predicted dependence on the molecular coverage (cf. Fig 4b.)

Revision

We have added new paragraphs to the manuscript: “The exciton-molecule coupling has been studied both theoretically and experimentally in carbon based 2D structures (Refs. 27,30), however, neither graphene nor carbon nanotubes showed the appearance of an additional peak in the absorption spectrum.” and “Besides the idea of how to activate dark exciton states with molecules, our work also presents a recipe of how to directly measure the dispersion relation of dark excitons by exploiting the remarkable dependence of optical spectra on the molecular coverage.”

Comment 3

As for their proposal for using this effect for sensing molecule, it is also not entirely clear how important this effect will be in a realistic experiment. [...] Can this scheme be ever used to detect a single molecule of reasonable dipole moment?

Reply

The detection of molecules based on the scheme used in our manuscript has already been applied in previous experimental studies on functionalized carbon nanotubes (E. Malic et al. "Carbon nanotubes as substrates for molecular spiropyran-based switches." *Journal of Physics: Condensed Matter* 24.39 (2012): 394006.) Here, the focus was on the conventional sensor schemes including peak shifts and intensity changes. Our current work reveals now the possibility of a new sensor mechanism that according to Fig. 4A should work for molecules exhibiting a dipole moment in the range of 10 Debye. One very prominent example for such molecules is the photoactive merocyanine molecule. As discussed in comment 9 of the referee #1, variation of substrate and temperature can be used to further optimize the required range of the dipole moments.

Revision

See the revision to comment 9 of the referee #1.

Comment 4

More importantly, I feel that the authors have overlooked certain mechanisms that can seriously undermine their scheme. For example, the relaxation of the exciton through non-radiative channels which are likely to open when the exciton brought in the vicinity of the molecule which has other degrees of freedom (vibrational, etc) and can not be simply modeled as a dipole.

Reply

The referee raises an important point. In this work, we have focused on non-covalent functionalization of TMDs with molecules, where the bonding is weak and happens via van der Waals interaction. For this functionalization scenario it is well known that the electronic structure of both systems remains unchanged to a large extent. Therefore, we also expect that the impact of additional relaxation channels will be small. This assumption is supported by above mentioned experiments performed on non-covalently functionalized carbon nanotubes, where no significant increase in excitonic linewidths through additional non-radiative channels has been observed.

We also agree with the referee that the exciton-molecule coupling will not only have an impact on the optical properties of the TMD, but will also affect optical and vibrational properties of the attached molecules. However, we expect that a change in the vibrational modes of the molecules will not significantly vary the molecular dipole moment or the molecular coverage, which are the crucial properties for the proposed sensor mechanism.

Revision

We have added a new paragraph to the manuscript: "Here, we focus on the impact of the exciton-molecule interaction on the optical properties of the TMD and neglect the changes in optical and vibrational properties of the molecules. We expect these effects to have a minor influence on the activation of dark states in TMDs, where the molecular dipole moment and the coverage play the crucial role."

Comment 5

The title "Out of the dark and into the light: ..." is probably suitable for a popular commentary but not for a regular scientific article.

Reply & Revision

As a response to the reviewer's comment we suggest an alternative title: "Proposal for dark exciton based chemical sensors."

Comment 6

The figures with gradient shading although look artistic, they don't convey any information and can unnecessary confuse the reader who is trying to gain a quantitative understanding.

Reply & Revision

To meet the criticism of the referee, we have emphasized the single lines in all figures. This way one can clearly follow the curves and one still has the color shading emphasizing the different temperatures or substrates.

Comment 7

The use of effective mass Wannier equation is not strictly valid for TMD excitons as was recently shown in two publications - Phys. Rev. Lett. 115, 166802 and 166803 (2015). Although for the current analysis, Berry curvature effects are likely to play a minor role, the shortcomings of the Wannier equation for TMDs should be mentioned.

Reply

We have not explicitly stated in our manuscript that Berry curvature effects are actually included in our approach within the tight-binding wave functions. Hence, the excitonic eigen energies and wave functions obtained by solving the effective mass Wannier equation do include the Berry phase. However, our calculations reveal that for the microscopic polarization of the bright and dark excitons (Eqs. (4), (5)) the phase does not play a role, since it cancels out in the exciton-molecule coupling element

$$G_{\mathbf{Q}\mathbf{k}}^{\mu\nu} = \sum_{\mathbf{q}} (\varphi_{\mathbf{q}}^{\mu*} g_{\mathbf{q}_{\alpha}, \mathbf{q}_{\alpha} + \mathbf{k}}^{cc} \varphi_{\mathbf{q} + \beta \mathbf{k}}^{\nu} - \varphi_{\mathbf{q}}^{\mu*} g_{\mathbf{q}_{\beta} - \mathbf{k}, \mathbf{q}_{\beta}}^{vv} \varphi_{\mathbf{q} - \alpha \mathbf{k}}^{\nu})$$

because $phase(g_{\mathbf{q}_{\alpha}, \mathbf{q}_{\alpha} + \mathbf{k}}^{cc}) = -phase(\varphi_{\mathbf{q}}^{\mu*} \varphi_{\mathbf{q} + \beta \mathbf{k}}^{\nu})$ and $phase(g_{\mathbf{q}_{\beta} - \mathbf{k}, \mathbf{q}_{\beta}}^{vv}) = -phase(\varphi_{\mathbf{q}}^{\mu*} \varphi_{\mathbf{q} - \alpha \mathbf{k}}^{\nu})$.

Revision

We have added to the manuscript: "Note that the tight-binding coefficients also include a non-zero geometric phase [35-37]. However, due to the symmetry of the exciton-molecule coupling, this phase cancels out for our studies."

Comment 8

In Fig. 4, what dipole density was assumed for the left panel and what dipole moment was assumed for the right panel?

Reply

The calculations are performed at a dipole density 0.8 nm^{-2} for the left panel and a dipole moment of 13 Debye for the right panel.

Revision

We have added a sentence to the manuscript to clarify the standard parameters: "If not otherwise stated we have used for the calculations WS2 on a silicon dioxide substrate at 77 K with attached merocyanine molecules with a dipole moment of 13 Debye and a molecular coverage of 0.8 nm^{-2} ."

Comment 9

Can the authors offer a simple explanation between the dark and bright exciton due to the presence of the dipole i.e., demystify $G_{\{0n\}^{\wedge}\{K\}} G_{\{0n\}^{\wedge}\{K\}}$ (see above)?

Reply

Bright excitons can be accessed directly with one-photon excitation. For dark excitons, there is an additional momentum transfer needed (indirect transitions). In our work, this momentum is induced by

the attached molecules. One can imagine a molecular lattice in the real space, where the corresponding lattice constant determines the center-of-mass moment that can be provided by the molecules for indirect scattering within the actual TMD lattice. By changing the molecular coverage, we can induce different indirect transitions and this way we can map the dispersion of the dark excitons. This is a simple picture and the concrete realization of the process strongly depends on the actual spectrum of bright and dark states in TMDs as well as on molecular characteristics, such as dipole moments and molecular coverage.

Revision

We have added an additional paragraph to the manuscript: “One can imagine a molecular lattice in the real space, where the corresponding lattice constant determines the center-of-mass moment that can be provided by the molecules for indirect scattering within the actual TMD lattice.”

REVIEWERS' COMMENTS:

Reviewer #1 (Remarks to the Author):

After considering most of all the comments/corrections proposed by the different referees, I think the manuscript could be suitable for publication, after a minor revision that do not require a further review. I would like to see in the present communication, the arguments proposed by the authors when considering the question raised in comment #3 of the referee #1.

If I understand correctly, the authors have disregarded K-K or K-K' dark excitons in their approach and considered only K-L excitons because of:

- (i) « the strongest exciton-molecule interaction due to the enhanced density of states in the L valley enhancing the overlap of the involved excitonic wave functions.»
- (i) K-L exciton is energetically below the bright K-K one.

It would be interesting to provide a clear evidence of the point (i) in the manuscript to help the reader understanding this issue and at the same time mentioned that in the case of W-based TMDCs dark K-K states are also below the bright one, as proposed in PRB 93 121107(R) for instance.

Reviewer #2 (Remarks to the Author):

The authors have suitably addressed my previous concerns, and I believe the paper can now be accepted for publication.

Reviewer#1

After considering most of all the comments/corrections proposed by the different referees, I think the manuscript could be suitable for publication, after a minor revision that do not require a further review.

We thank the referee for the positive evaluation of our work.

Comment 1

I would like to see in the present communication, the arguments proposed by the authors when considering the question raised in comment #3 of the referee #1. If I understand correctly, the authors have disregarded K-K or K-K' dark excitons in their approach and considered only K-L excitons because of: (i) « the strongest exciton-molecule interaction due to the enhanced density of states in the L valley enhancing the overlap of the involved excitonic wave functions.» (ii) K-L exciton is energetically below the bright K-K one. It would be interesting to provide a clear evidence of the point (i) in the manuscript to help the reader understanding this issue and at the same time mentioned that in the case of W-based TMDCs dark K-K states are also below the bright one, as proposed in PRB 93 121107(R) for instance.

Response &Revision

We have added a sentence to the methods section addressing the strength of the exciton-molecule coupling element for the different dark excitons (KL, KK, KK'). Moreover, we have added a reference (Ref. 15) to the main text to mention the existence of spin-forbidden dark excitons.

Reviewer#2

The authors have suitably addressed my previous concerns, and I believe the paper can now be accepted for publication.